# Evaluation of Rhesus Macaque Models for Cerebral Palsy

**DOI:** 10.3390/brainsci12091243

**Published:** 2022-09-14

**Authors:** Yong Zhu, Yanan Xiong, Jin Zhang, Haiyang Tong, Hongyi Yang, Qingjun Zhu, Xiaoyan Xu, De Wu, Jiulai Tang, Jinhua Li

**Affiliations:** 1School of Life Sciences, Hefei Normal University, No. 1688, Lianhua Road, Hefei 230601, China; 2High Magnetic Field Laboratory, Chinese Academy of Sciences, No. 350, Shushanhu Road, Hefei 230031, China; 3Pediatric Neurological Rehabilitation Center, the First Affiliated Hospital of Anhui Medical University, No. 218, Jixi Road, Hefei 230022, China; 4Department of Emergency, Tongji Hospital of Tongji University, No. 389, Xincun Road, Shanghai 200065, China; 5International Collaborative Research Center for Huangshan Biodiversity and Tibetan Macaque Behavioral Ecology, School of Resources and Environmental Engineering, Anhui University, No. 111, Jiulong Road, Hefei 230601, China

**Keywords:** cerebral palsy, rhesus macaque, animal model, evaluation methodology

## Abstract

**Simple Summary:**

Cerebral palsy (CP) is caused by a non-progressive injury that affects the central nervous system during early development. According to the World Health Organization, CP occurs in approximately 2–4 of 1000 newborns. Animal models, especially non-human primate models, have greatly assisted our understanding of CP, and have provided simulative objects for treatment. In this study, the rhesus macaque CP models were established by partial resection of the motor cortex and the intrathecal injection of bilirubin. We evaluated abnormal posture, motor dysfunction, gross and fine motor behavior, muscular tension of rhesus macaque CP models, and changes in the cerebral cortex and basal ganglia, the last of which were observed using magnetic resonance imaging. This model evaluation methodology may guide researchers through the model building process. The findings serve as a reference for establishing and assessing non-human primate CP models.

**Abstract:**

Animal models play a central role in all areas of biomedical research. The similarities in anatomical structure and physiological characteristics shared by non-human primates (NHPs) and humans make NHPs ideal models with which to study human disorders, such as cerebral palsy (CP). However, the methodologies for systematically evaluating NHP models of CP have rarely been assessed, despite the long history of using NHP models to understand CP. Such models should be evaluated using multidisciplinary approaches prior to being used to research the diagnosis and treatment of CP. In this study, we evaluated rhesus macaque CP models established by partial resection of the motor cortex and intrathecal injection of bilirubin. Abnormal posture, motor dysfunction, gross and fine motor behavior, and muscular tension were evaluated, and changes in the cerebral cortex and basal ganglia were observed using 9.4 T magnetic resonance imaging. The results clearly demonstrated the utility of the established evaluation methodology for assessing CP models. This model evaluation methodology may guide researchers through the model building process.

## 1. Introduction

Cerebral palsy (CP) is caused by a non-progressive injury that affects the central nervous system during early development [1,2]. According to the World Health Organization, CP occurs in approximately 2–4 of every 1000 newborns [3]. CP consists of a set of disorders including postural deficits, motor dysfunctions, and cognitive and mobility difficulties [1,2]. These disorders severely hinder the patient’s performance in functional activities such as feeding, locomotion, dressing, personal care, and social interaction, thus seriously affecting their daily life [4]. Approximately 80–95% of CP cases develop during the prenatal period, and only a small number of cases develop during and after birth. However, sometimes cerebral palsy is caused by postnatal factors, extracranial injury, or bilirubin encephalopathy. The incidence of physiologic jaundice in neonates in China is 60–70%, almost all of which are benign. The incidence of Kernicterus is about 4.8%. Kernicterus is characterized by the yellow staining of the brain regions that are susceptible to bilirubin staining (including the basal nucleus, subthalamic nucleus, globus pallidus, subthalamic nucleus, brainstem cerebellum) and other neural nuclei. This type of brain injury results in dyskinetic cerebral palsy with extrapyramidal signs typical of dance athetosis and/or dystonia. Few existing studies have reported animal models of this type. The mechanisms underlying the occurrence of the causative injury and the development of CP remain unknown. Since animal models have greatly assisted in understanding CP and have provided simulative objects for treatment [1,5,6], it is also important to develop an animal model of CP that represents the pathophysiology leading to CP.

Models of CP range from small to large animals. However, owing to their physiological, anatomical, and behavioral similarities to humans, non-human primates (NHPs) are ideal candidates for safe and efficient preclinical tests for neurotherapeutic interventions [7,8]. If NHP models that closely mimic the neuropathological damage of CP can be established, then the underlying cause of CP may be uncovered [9]. Creating an NHP model with injury at the correct stage of brain development is vital in order to obtain consistent results; then, neurodevelopmental tests adapted for humans can be applied [6].

Some studies used hypoxia and preterm methods to establish NHP CP models [10,11]. Near-total birth asphyxia was used in neonatal rhesus macaques (*Macaca mulatta*) and pigtailed macaques (*Macaca nemestrina*) to study the effects of hypoxia as a cause of neurological deficits [5,12,13,14]. Myers administered oxytocin (Pitocin) to female rhesus macaques to establish preterm CP models [15]. A baboon (*Papio papio*) model of CP was reported, for which baboons were prematurely delivered by hysterotomy at 125 days (of their 184-day gestational period) and cared for in an NICU in a similar way to preterm human infants [16]. Some studies have reported that bilirubin injection was used to produce animal models of cerebral palsy in animals such as rabbits and rats, models which have been widely accepted by scientists, but relevant experiments have not been conducted on NHP models. The authors directly established a model of cerebral palsy in macaques. We used six-month-old macaques to simulate the causes of cerebral palsy in humans. Partial excision of the motor cortex was used in one model and intrathecal bilirubin injection was used in the other. An “Animal model of cerebral palsy prepared by intrathecal injection of bilirubin in macaques” was patented by the State Intellectual Property Office (201611225803.2). Rhesus monkeys are very close to humans in body contour and genes, and are the best experimental NHPs for animal models of human diseases, so the establishment of a rhesus cerebral palsy model is superior to other animal models.

Despite the long history of using NHP models in the investigation of CP, the process of model evaluation has rarely been assessed systematically. Common neuropathologies in these models include reactive astrogliosis, the activation of microglia in white matter, and enlarged ventricles. In addition, significant brain injury was observed using magnetic resonance imaging (MRI) scans [17]. Methods for evaluating NHP CP models are not comprehensive and tend to focus on neuropathologies without evaluating behavior. This is possibly because behavioral indicators are not usually objective or because simpler and more scientific evaluation methods are more intuitive and efficient [18]. Given the importance of evaluating models prior to using them for diagnostic and treatment research, we developed an improved method for systematically evaluating models that included assessing behavior. Our main objective was to provide a new method for evaluating the rhesus macaque CP model and to help future researchers develop more effective models.

## 2. Materials and Methods

### 2.1. Subjects

We established four rhesus macaque CP models. One model was established by the partial resection of the motor cortex (CP model 1). The cortex of the precentral gyrus and the posterior superior frontal gyrus (approximately 0.3–0.5 cm the anterior middle sulcus) was excised the lateral fissure of the right brain to the interhemispheric fissure, to a depth of approximately 0.5–0.6 cm. The other three models (CP models 2–4) were established by the intrathecal injection of bilirubin. The macaque was placed on the operating table on its side, with its back perpendicular to the surface of the bed. The head was bent forward toward the chest, so that the trunk was arched and the spine was kyphotic, in order to widen the intervertebral space. The intervertebral disc space two, below the intersection of the iliac crest line and the posterior median line, was taken as the puncture point. After routine disinfection of the skin, sterile gloves were worn, and the body was covered with the disinfection towel. Local infiltration with 2% lidocaine was used for anesthesia. The left hand was used to position the skin, the right hand was used to hold the puncture needle vertically, and the needle tip was slightly inclined in the direction of the head. When a breakthrough was felt twice, the needle core was slowly withdrawn until cerebrospinal fluid outflow could be seen. The depth of the needle was 2–3 cm. The bilirubin solution was added, the syringe wrapped with tin foil, and the solution injected at 10 ug/g, while maintaining the injection rate at 1 uL/min. The above procedures were performed while the macaques were anesthetized to minimize the animals’ suffering. Normal macaques of the same age which had not undergone experimental procedures served as the control group. All macaques were operated on at 6 months of age at the High Magnetic Field Laboratory, Chinese Academy of Sciences. The macaques were provided by the Anhui ShengPeng Experimental Animal Technology Co. Ltd., Anhui, China.

### 2.2. Qualitative Evaluation

The characteristics and motor dysfunctions of the CP models, including abnormal posture, were evaluated. The approach allowed for intuitive, quick, and easy assessment. Abnormal posture was assessed by observing whether the model animals presented a supine position or prone position at rest, and whether the body leaned to one side, with limbs bucked, or presented an asymmetric posture when squatting. Motor dysfunction was assessed by observing whether the model animals presented abnormal motor function, including the body leaning to the left or right, dyskinesia, and motor incoordination.

### 2.3. Quantitative Evaluation

Behavior in the CP models was assessed based on a behavioral scale and grading criteria, which included gross motor behavior, fine motor behavior, and muscular tension. This evaluation provided an objective and highly reliable assessment. Gross motor behavior included standing, moving, running, jumping, hanging, hanging upside down, climbing, and expression of fear. Each behavior was divided into the following three ranks: high was assigned 2 points, middle 1 point, and low 0 points (full score 16 points; see Table 1). The higher the score, the stronger the gross motor function. Fine motor behavior included licking, pinching, feeding, handling, and grooming. Each behavior was divided into the same three ranks (full score 10 points: see Table 2). The higher the score, the stronger the fine motor function. To assess muscular tension, we adopted the modified Ashworth scale, and freedom degrees of motor and joint flexion–extension were divided into Grades 0–5 (Table 3). The higher the score, the higher the muscle tone.

### 2.4. Imaging Evaluation

Imaging studies were performed at the Magnetic Resonance Imaging Laboratory using a high-field MRI (9.4T/Φ400, 9.4-tesla magnet and 400 mm clear bore). Changes in cerebral imaging were detected by 9.4T MRI.

### 2.5. Statistical Analysis

SPSS 17.0 statistical software was used for statistical analysis, data were compared through a rank sum test in the compatibility group, and a multiple-comparison among multiple samples was performed. *p* < 0.05 was considered to indicate significance.

## 3. Results

### 3.1. Abnormal Posture

Macaques in the operating group showed abnormal postures at rest. CP model 1 often squatted in the corner of the cage, with the left upper limb placed on the cage to maintain balance, and the left lower limb presenting a buckling state of contraction. The left and right sides of the body were not symmetrical. There was no evident change in symptoms between 1- and 2-year-old macaques. CP models 2–4 showed a supine or prone position at rest. The front and hind legs all presented buckling. There was no evident change in the symptoms between 1- and 2-year-old macaques (Figure 1).

### 3.2. Motor Dysfunction

Postoperative left limb motor dysfunction occurred in the CP model following partial resection of the motor cortex. This model presented left hemiplegia; feeding and other activities were mainly carried out with the right limb, and the model was in the prone position at rest. After 6 weeks, the macaque sat permanently in the corner of the cage and was much less active than the normal control group. The CP models undergoing intrathecal injection of bilirubin remained in a supine or prone position, and were significantly less active than the normal control group.

### 3.3. Gross and Fine Motor Behavior

Compared with the normal macaque, the score for gross motor behavior began to decrease 1 month after the operation, and showed a trend of decreasing until 12 months after the operation (*χ*^2^ = 11.086, *df* = 4, *p* = 0.026) (Figure 2). Compared with the normal macaque, the score for fine motor behavior began to decrease 1 month after the operation, and continued to do so until 6 months after the operation (*χ*^2^ = 12.571, *df* = 4, *p* = 0.014) (Figure 3).

### 3.4. Muscular Tension

Compared with the normal macaque, the muscular tension of all CP models increased (*χ^2^* = 11.451, *df* = 4, *p* = 0.022). All models showed an increasing trend following the operation. The highest score for muscular tension was 4 points at 12 months, and this state was maintained in the following month (Figure 4).

### 3.5. MRI

Compared with the normal macaques, changes after partial resection of the motor cortex, as well as scar tissue formation, were observed in the cerebral MRI of macaques 3 weeks after the operation. From the MRI, it was clear that the basal ganglia, hippocampus, and sub thalamic nucleus had been damaged by bilirubin (Figure 5).

## 4. Discussion

In this study, we evaluated abnormal posture, motor dysfunction, gross and fine motor behavior, and muscular tension of rhesus macaque CP models. Changes in the cerebral cortex and basal ganglia were also observed using MRI. The results clearly demonstrated the utility of the established evaluation methodology for assessing CP models. This model evaluation methodology may guide researchers through the model building process.

For behavior evaluation, more behavioral indicators could be added, such as barrier-crossing behavior, aggressive behavior, repetitive behavior, and some reflex behaviors. In this study, the neural reflex of the macaques was not tested since it is difficult to elicit, and there are no relevant neural reflex scores available. The frequently used behavior evaluation methods for human CP are the Neonatal Behavioral Neurological Assessment (NBNA) and Neonatal Gross Motor Function Measure scales (GMFM-88 and GMFM-66) [19,20]. The human behavior assessment was referenced and combined with behaviors specific to macaques. To achieve fast and effective behavioral evaluation, common and classical motor behaviors were used as our evaluation indicators.

The imaging techniques of MRI, such as arbitrary-place imaging, high resolution of soft tissue, and diffusion tensor imaging, contribute to pathological research on CP models [21,22]. These techniques were used to detect cranial changes before and after the operation, and provided specific information on the nature, site, and scope of brain lesions [17,23]. The gradually improved spasm symptom at the later stage is related to the protective contralesional hemispheric compensation in macaques, which also accounts for the unsustainable symptoms. Postoperative injury sites in macaques could still be seen in MRI images at the later stage, but there were no other evident changes.

The anatomical structure and physiological characteristics of NHPs are close to those of humans, making NHPs better models than small–middle-sized animals. With the demands of clinical needs, more NHPs, such as macaques, have been used to establish models for CP or other diseases. Therefore, it is important to develop a systematic method to evaluate rhesus macaque CP models. More importantly, the model evaluation may guide researchers through the model building process.

## 5. Conclusions

As our results clearly demonstrate, the described evaluation method can systematically evaluate rhesus macaque CP models, and includes behavioral evaluation, morphological characterization, and MRI observation. Furthermore, this evaluation method can quantify several specific behaviors of macaques that are usually assessed with clinical rating scales, such as gross and fine motor behaviors. In addition to evaluating the change or improvement of specific activity behaviors after treatment, this method can also evaluate the effectiveness of the therapeutic strategy used. Our study provides insights into improving the evaluation of such models; however, further improvement and refinement will be needed to enhance and expand diagnostic and therapeutic research.

## Figures and Tables

**Figure 1 brainsci-12-01243-f001:**
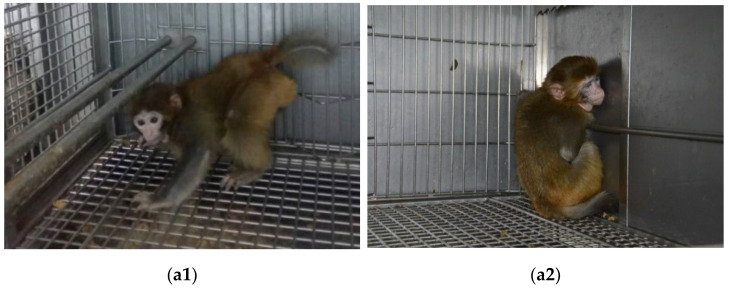
Abnormal posture in 1- and 2-year-old macaques. (**a1**) Normal control (1 year); (**a2**) CP by partial resection of motor cortex; (**a3**) CP by intrathecal injection of bilirubin (1 year); (**a4**) CP by intrathecal injection of bilirubin (1 year); (**b1**) normal control (2 years); (**b2**) CP by partial resection of motor cortex (2 years); (**b3**) CP by intrathecal injection of bilirubin (2 years); (**b4**) CP by intrathecal injection of bilirubin (2 years).

**Figure 2 brainsci-12-01243-f002:**
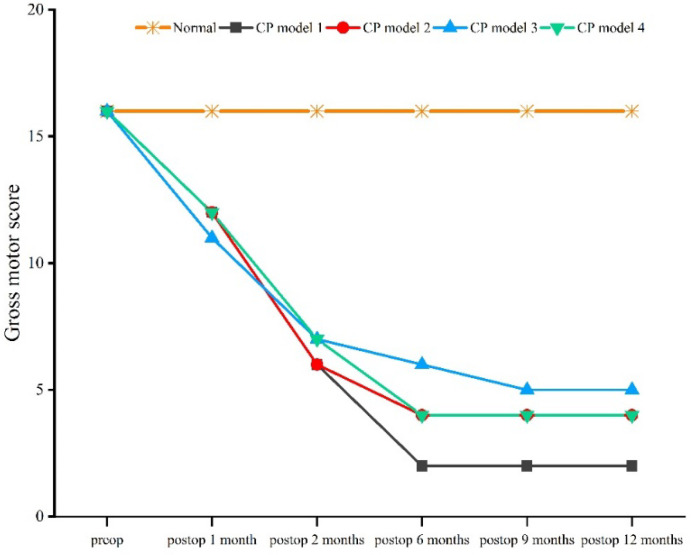
Gross behavior score.

**Figure 3 brainsci-12-01243-f003:**
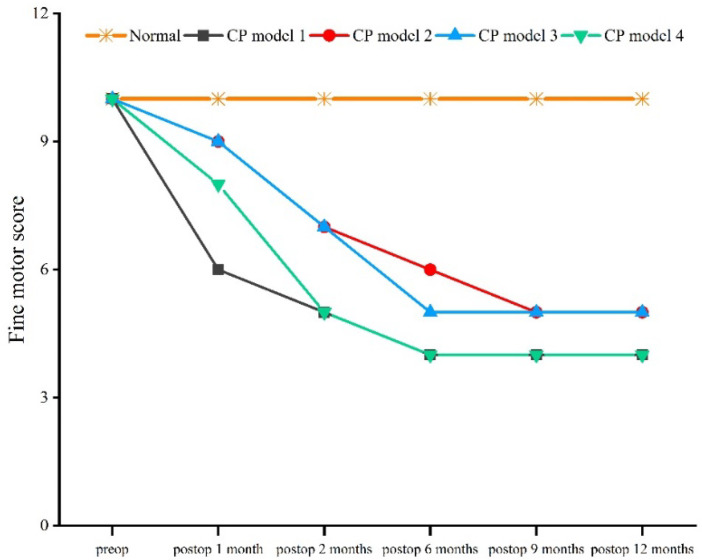
Fine behavior score.

**Figure 4 brainsci-12-01243-f004:**
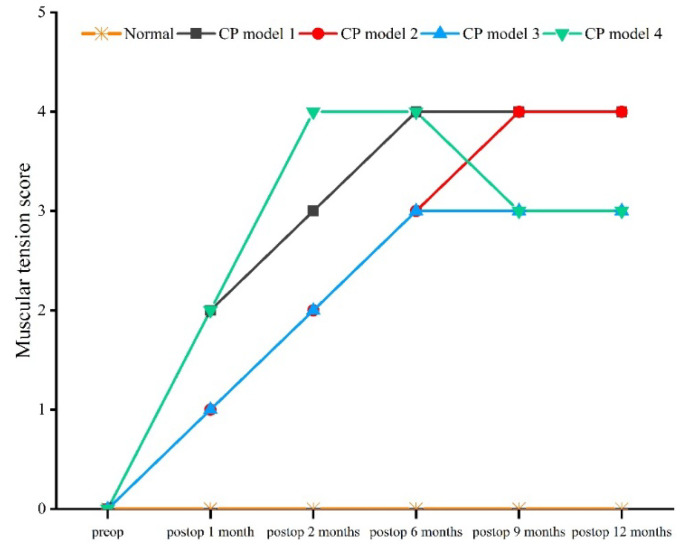
Muscular tension score.

**Figure 5 brainsci-12-01243-f005:**
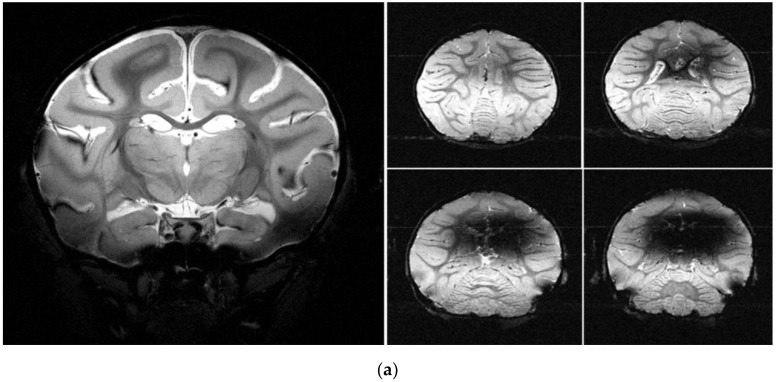
MRI of CP models. (**a**) Left: normal macaque; right: CP by intrathecal injection of bilirubin; (**b**) the arrows point to the location of the globus pallidus in the basal ganglia; (**c**) left: normal macaque; right: CP by partial resection of motor cortex (the arrows in the right panel of (**b**) point to the brain areas damaged by partial resection of the motor cortex).

**Table 1 brainsci-12-01243-t001:** Gross motor behavior and scoring standards.

Behavior	Scoring Standards	Score
Standing	Unable to stand	0
	Standing unstable	1
	Standing stable and sustainable	2
Moving	Unable to move	0
	Disturbance of gait and coordination	1
	Moving smoothly	2
Running	Unable to run	0
	Unstable running	1
	Fast complete movement	2
Jumping	Unable to jump	0
	Take-off posture	1
	Jumping	2
Hanging	Unable to hang	0
	Grasping but unable to hang	1
	Hanging and shaking the body	2
Hanging upside down	Unable to hang upside down	0
	Lasts less than 3 s	1
	Hanging upside down	2
Climbing	Unable to climb	0
	Relying on cage or short distance for climbing	1
	Climbing on the cage	2
Fearful behavior	Body without reaction when stimulated	0
	Body appearing to move when stimulated	1
	Body moving smoothly when stimulated	2

**Table 2 brainsci-12-01243-t002:** Fine motor behavior and scoring standards.

Behavior	Scoring Standards	Score
Lick	Unable to lick a 10 cm^2^ board with cream	0
	Licking half the board	1
	Licking the board clean	2
Pinch	Toe unable to bend	0
	Unable to successfully pinch objects	1
	Complete pinch movement	2
Feeding	Unable to finish feeding action	0
	With feeding movement, without eating	1
	Complete feeding movement	2
Handle	Unable to handle things	0
	Unable to maintain handling position	1
	Complete handling movement	2
Grooming	Unable to groom itself	0
	Grooming actions not coherent	1
	Grooming coherent and sustained	2

**Table 3 brainsci-12-01243-t003:** Grades of muscular tension and scoring standards.

Grade	Scoring Standards	Score
0	With no increased muscle tone, but could move freely	0
1	With slightly increased muscle tone, and no resistance in the affected limb for passive activity within the entire scope	1
2	With slightly increased muscle tone, the former 1/2 ROM * of the affected limb was slightly stuck in passive activity, while the latter 1/2 ROM showed slight resistance	2
3	With mildly increased muscle tone, the affected limb showed resistance within most of ROM in passive activity, but could still move	3
4	With moderately increased muscle tone, the affected limb showed resistance within the entire ROM in passive activity, and activity was difficult	4
5	With highly increased muscle tone, the affected limb was stiff, showing great resistance, and passive activity was extremely difficult	5

* note: ROM means range of motion.

## Data Availability

The data presented in this study are available in the manuscript.

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
