# Peer review of "Evaluation of Rhesus Macaque Models for Cerebral Palsy"

_brainsci, 2022, doi:10.3390/brainsci12091243_

Round 1
Reviewer 1 Report
The manuscript entitled “Evaluation of rhesus macaque models for cerebral palsy” by Zhu and coworker is focusing the actual topic and deserves special attention for its actuality. The manuscript describes evaluation method that can systematically evaluate rhesus macaque CP models and covers behavioral evaluation, morphological characterization, and MRI observation. The species employed in this study is extremely rare and therefore presented results are especially valuable and respectable.
Statistical significance should be indicated on the graphs.
The authors should pay special attention to the technical preparation of figures since Figures 2 and 4 are not clearly presented.
Check spelling and grammar errors.
Reviewer 2 Report
The presented work is of great interest due to complex approaches developed by clear algorithms and an ordered system for quantifying and tomographic diagnosis of a new model of cerebral palsy in the rhesus macaque. The work contains substantiated statements and a clear statement of the problem, which is of significant practical importance, which determines the high relevance of this study. The Introduction section is quite logical and justified and leaves a clear idea of ​​the problem stated by the authors. The methodological part is also quite sufficient, and allows to reproduce individual fragments of experimental manipulations and control the results. In the productive part of the article, however, it is necessary to more fully present the stages of the bilirubin effect on the centers of the brain and give a more complete description of exactly how intracerebral administration of bilirubin destroys the centers of the brain: the basal ganglia, the hippocampus and the subthalamic nucleus. It is also necessary to indicate whether this model of cerebral palsy was developed directly by the authors, or whether it is generally accepted on other animal models. A broader discussion of the model presented by the authors with existing models on other animals is also needed. On the whole, the work deserves high praise and, after making the necessary clarifications, it can be recommended for publication in Brain Science.
Reviewer 3 Report
This paper establishes two NHP models of cerebral palsy by partial resection of the motor cortex and intrathecal injection of bilirubin. However, one of my biggest concerns is if these models are best for studying cerebral palsy or for studying other types of acute brain damage.
Please address the following questions:
1) The author mentioned that “Approximately 80–95% of CP cases develop during the prenatal period, and only a small number of cases develop during and after birth.” However, the monkeys used in this study were induced cerebral palsy at the age of 6 month. So, how this rhesus macaque models could be appropriately used for study human CP cases?
2) Were the experiments double-blind? Did the experimenter know if the monkeys were from control group or CP group when scoring them?
3) Figures 2-4 did not have data from the control group. Can the author please provide them?
4) How long was per evaluation session? Did the authors only evaluate one session at per time point?
5) Line 171-172, can the authors provide how the comparison was performed?
6) For all the figures, please increase the font size so that they are more visible to readers.
7) Figure 6, please describe what the arrows in right panel of b point to. In the caption of Figure 6, it says right b is the MRI from the bilirubin monkey. However, how did bilirubin only cause brain damage on the left hemisphere? Should Figure 6a be the bilirubin monkey and Figure 6b be the motor cortex recession monkey? Please clarify.
8) In Figure 6, please add scalar bar. Also, can the authors add arrows to basal ganglia, hippocampus, and sub thalamic nucleus that were damaged by bilirubin?
Round 2
Reviewer 3 Report
The authors have resolved my concerns. Thanks!